# Comparison of Heart Autonomic Control between Hemodynamically Stable and Unstable Patients during Hemodialysis Sessions: A Bayesian Approach

**DOI:** 10.3390/e25060883

**Published:** 2023-05-31

**Authors:** Natália de Jesus Oliveira, Alinne Alves Oliveira, Silvania Moraes Costa, Uanderson Silva Pirôpo, Mauro Fernandes Teles, Verônica Porto de Freitas, Dieslley Amorim de Souza, Rafael Pereira

**Affiliations:** 1Integrative Physiology Research Center, Department of Biological Sciences, Universidade Estadual do Sudoeste da Bahia, Jequie 45210-506, BA, Brazil; 2Research Group in Neuromuscular Physiology, Department of Biological Sciences, Universidade Estadual do Sudoeste da Bahia, Jequie 45210-506, BA, Brazil; 3Medicine School, Universidade Estadual Do Sudoeste da Bahia (UESB), Jequie 45210-506, BA, Brazil; 4Postgraduate Program in Nursing & Health, Universidade Estadual do Sudoeste da Bahia, Jequie 45210-506, BA, Brazil; 5Faculdade Santo Agostinho (FASA), Vitória da Conquista 45028-100, BA, Brazil

**Keywords:** chronic kidney disease, heart rate variability, hemodynamics

## Abstract

Intradialytic hypotension is a common complication during hemodialysis sessions. The analysis of successive RR interval variability using nonlinear methods represents a promising tool for evaluating the cardiovascular response to acute volemic changes. Thus, the present study aims to compare the variability of successive RR intervals between hemodynamically stable (HS) and unstable (HU) patients during a hemodialysis session, through linear and nonlinear methods. Forty-six chronic kidney disease patients volunteered in this study. Successive RR intervals and blood pressures were recorded throughout the hemodialysis session. Hemodynamic stability was defined based on the delta of systolic blood pressure (higher SBP-lower SBP). The cutoff for hemodynamic stability was defined as 30 mm Hg, and patients were stratified as: HS ([n = 21]: ≤29.9 mm Hg) or HU ([n = 25]: ≥30 mm Hg). Linear methods (low-frequency [LFnu] and high-frequency [HFnu] spectra) and nonlinear methods (multiscale entropy [MSE] for Scales 1–20, and fuzzy entropy) were applied. The area under the MSE curve at Scales 1–5 (MSE_1–5_), 6–20 (MSE_6–20_), and 1–20 (MSE_1–20_) were also used as nonlinear parameters. Frequentist and Bayesian inferences were applied to compare HS and HU patients. The HS patients exhibited a significantly higher LFnu and lower HFnu. For MSE parameters, Scales 3–20 were significantly higher, as well as MSE_1–5_, MSE_6–20_, and MSE_1–20_ in HS, when compared to HU patients (*p* < 0.05). Regarding Bayesian inference, the spectral parameters demonstrated an anecdotal (65.9%) posterior probability favoring the alternative hypothesis, while MSE exhibited moderate to very strong probability (79.4 to 96.3%) at Scales 3–20, and MSE_1–5_, MSE_6–20_, and MSE_1–20_. HS patients exhibited a higher heart-rate complexity than HU patients. In addition, the MSE demonstrated a greater potential than spectral methods to differentiate variability patterns in successive RR intervals.

## 1. Introduction

The low glomerular filtration rate, the main parameter used to diagnose chronic kidney disease (CKD) [1], was associated with 2.2 million deaths worldwide in 2013 [2]. Hemodialysis is the main renal replacement therapy for CKD patients [3], but many complications are observed during the hemodialysis sessions, such as nausea, headache, chest pain, dizziness, and hypotension, among others [4,5,6].

Intradialytic hypotension (IH) is the most frequent and dangerous complication during the hemodialysis session, requiring immediate attention [7]. In fact, IH is indicated as a predictor of increased mortality in dialysis patients [8,9], owing to the predisposition to myocardial ischemia, cardiac arrhythmia, and increased risk of sudden death cardiac arrest [10].

IH is triggered by many factors, the intravascular volume depletion, associated with reduced osmolality, as occurs in patients undergoing a high rate of ultrafiltration, favors IH occurrence [11]. Previous diseases such as heart and circulatory diseases, as well as diabetes and autonomic dysfunctions, may also contribute [11]. Indeed, the autonomic nervous system (ANS) is the major contributor to immediate cardiovascular adjustments to acute intravascular volume changes. Thus, understanding the autonomic adjustments during hemodialysis sessions, as well as the trends of these adjustments from different CKD patients, may help to plan preventive and/or therapeutic strategies to deal with IH.

Heart-rate variability (HRV) analysis stands out as a non-invasive and low-cost assessment approach to investigate the ANS influence over the cardiovascular system, which is performed by measuring changes (i.e., variability) in successive RR intervals [12], supporting the concept of heart autonomic control. Nonlinear methods have been used as a promising approach to investigate HRV and have been reported as more suitable to extract information from complex systems, as well as better than linear methods to predict many clinical outcomes [13,14].

Among the nonlinear methods, entropy measurements have been pointed out as a promising way to deal with high-complexity biomedical data [15]. In particular, a recent approach called multiscale entropy (MSE) seems to be a promising tool for many applications such as cardiovascular diagnosis, risk stratification, drug toxicity detection, and others [16]. With the MSE approach, the entropy of a given signal is quantified at various time scales, allowing the investigation of complex systems, which are characterized by mechanisms of physiological control occurring at multiple levels (i.e., subcellular to systemic levels) and operating over multiple time scales [16,17].

Costa et al. [16] presented a consistent explanation regarding the limitations involved in the use of single scale-based measures of entropy, and the potential of multiscale ones to access complexity from biological data, as the successive RR intervals. Briefly, the continuous interaction among different regulatory systems (i.e., multiple levels and time scales) ensures that this information is constantly transmitted at all these organizational levels, allowing the system to make adjustments according to a continuously changing environment [16,18].

As biological systems with higher entropy exhibit a greater adaptive capacity [19], we hypothesized that a greater entropy of successive RR interval variability could be associated with a better hemodynamic stability during a hemodialysis session. Therefore, the present study aimed to compare the variability of successive RR intervals between hemodynamically stable (HS) and unstable (HU) CKD patients during a hemodialysis session, through linear and nonlinear methods.

## 2. Methods

### 2.1. Sample

The sample consisted of 46 patients (49.0 ± 18.4 years old) diagnosed with CKD, and undergoing hemodialysis treatment for at least 6 months. Uncontrolled hypertension (systolic blood pressure (SBP) ≥200 mm Hg and/or diastolic blood pressure (DBP) ≥120 mm Hg) and a previous diagnosis of cardiac arrhythmia were the exclusion criteria. Volunteers were invited to take part in the study and were informed about the study procedures; those who agreed signed their informed consent. The study was approved by the Research Ethics Committee of the Universidade Estadual do Sudoeste da Bahia (protocol: 09635912.3.0000.0055).

### 2.2. Proceedings and Measurements

#### Blood-Pressure Recordings and Hemodynamic Stability Characterization

Blood-pressure measurements were carried out with automatic blood-pressure monitor HEM-OMRON^®^ 742INT. The systolic (SBP) and diastolic (DBP) blood pressures were recorded during each hemodialysis session. During the sessions, the recordings were performed at the beginning and at every 30 min during the hemodialysis session as we had performed previously [20]. As the sessions lasted for approximately 4 h, a total of eight blood-pressure measurements were taken. The interval between the penultimate and the last blood-pressure measurement was smaller than 30 min when the session lasted for less than 4 h.

Hemodynamic stability was established by the delta (i.e., the difference between the highest and lowest value) of the SBP from the eight records obtained during the hemodialysis session, as performed by Mascarenhas et al. [20].

The median of the SBP delta was used as the cutoff point to group the volunteers according to the hemodynamic stability level. Thus, the volunteers were assigned into two groups: the HD group (n = 21: SBP delta ≤ 29.9 mmHg) and the HU group (n = 25: SBP delta = or ≥30 mmHg). Considering the absence of consensus on the criterion to determine IH, we chose to stratify the studied sample according to the delta of SBP from our own sample, reaching the cutoff point of 29 mmHg, which is a value between the two most common criteria (i.e., 20 and 30 mmHg) used in previous studies [21,22,23]. Table 1 presents the sample characteristics and the filtered volume of the analyzed hemodialysis session.

### 2.3. RR Interval Recording

All patients who met the sample selection criteria and agreed to participate were subjected to the recording of RR intervals through a heart-rate monitor (Polar^®^ RS800CX, Kempele, Finland). Heart-rate monitors were previously validated for analysis of heart autonomic control [24] and exhibited good reliability in recording RR intervals during hemodialysis sessions [20,25]. The RR interval data were analyzed in the time and frequency domain and with MSE analysis, aiming to quantify the entropy from the successive RR interval dataset.

The recorded data were pre-processed using a smoothing procedure with a cutoff frequency of 0.035 Hz to remove disturbing low-frequency baseline trend components, as used and suggested by Luque-Casado et al. [26] and used by Silva et al. [25]. The procedures followed the recommendations of the Task Force of the European Society of Cardiology and the North American Society of Pacing and Electrophysiology [27], and all analyses were performed with the Kubios HRV analysis software 2.2 (Department of Applied Physics, University of Eastern Finland, Kuopio, Finland) [28].

### 2.4. Frequency Domain Analysis

The frequency domain analysis was performed using the fast Fourier transform (FFT) to obtain the following parameters: normalized magnitude from the spectrum of the low-frequency components (LFnu) and the high frequency (HFnu) components, and the LF/HF ratio. The low- and high-frequency power were set as 0.04–0.15 and 0.15–0.4 Hz, respectively. The window width was set to 256 s and overlapped to 50%, as we had performed previously [25].

### 2.5. Entropy Analysis

As entropy-based metrics, we obtained the MSE and fuzzy entropy (FuzzyEn). MSE is an extension of the entropy analysis used to obtain the parameter called sample entropy (SampEn), which incorporates two procedures [19]: (i) the original time series is fractionated into time series without overlapping points (i.e., successive RR intervals) and with increasing length (τ), where τ represents the scalar factor and which was selected to vary from 1 to 20; (ii) the time series were then submitted to the entropy analysis to generate entropy values for the 20 scales (scalar factor τ = 1 to 20), where τ = 1 represents the SampEn value of the original time series set. Aiming to obtain a global MSE index, we measured the area under the MSE curve constituted by entropy values on Scales 1–20 (MSE_1–20_) [29].

We also calculated the area under the MSE curve between 1–5 (MSE_1–5_) and 6–20 (MSE_6–20_) to represent the complexity between the short and long scales, respectively [30,31,32]. The area under the MSE curve between the cited scales may represent quantitative characteristics of the underlying physiological mechanisms at specific time scales (e.g., the area of Scale 1–5 (low) may correspond to the predominance of parasympathetic control, while the area of Scale 6–20 (high) may correspond to the predominance of sympathetic control) [30].

As the SampEn, FuzzyEn also estimates the average information carried by each sample of the signal but uses a different method to compute similarity patterns. In SampEn, two patterns are considered similar when the maximum difference between their corresponding points (d) is greater than the tolerance factor (r). While in FuzzyEn, a pair of patterns is not simply classified as similar or not. Instead, the distance between them (d) is passed through a fuzzy membership function to calculate the degree of similarity between them, as described by Silva et al. [33].

We obtained FuzzyEn measures using the software PyBioS, which uses the following fuzzy membership function: exp (−0.6931 × (d/r)^n^) [33].

### 2.6. Statistical Analysis

Continuous variables are presented as the mean ± standard deviation. Inferential statistics were performed using two approaches: frequentist inference and Bayesian inference.

A mixed linear model was applied to compare the linear (i.e., LFnu, HFnu) and nonlinear (i.e., MSE) parameters of the successive RR intervals’ variability between HS and HU CKD patients, with groups (i.e., HS vs. HU) as a fixed factor and the session filtered volume and uremic status as random factors. The session filtered volume and the uremic status were used to adjust the studied variables, since hyperuremia, a condition commonly observed in CKD patients, and the filtered volume can directly influence the sympathovagal modulation.

The results are presented as the mean ± SD, mean difference between groups, and its 95% confidence interval (95% CI). The mean differences and their 95% CI values were reported and interpreted as a measure of the effect size, since this approach allows the identification of the direction and magnitude of the effect, justifying its use as an adequate effect size measure [34], as used by Dos Santos et al., [35]. All frequentist procedures were carried out in SPSS 21.0 (IBM-SPSS Inc., Chicago, IL, USA), and the significance level was set as *p* ≤ 0.05.

Bayes factor hypothesis testing analyses were used to check the qualitative outcomes and the probability of replicating the same results (i.e., the magnitude of the evidence) [36]. Individual comparisons were based on the default *t*-test with a Cauchy (0, r = 1/sqrt) prior. The “U” in the Bayes factor denotes that it is uncorrected [37]. The outcomes were classified as anecdotal (BF10 = 1 to 3), moderate (3 to 10), strong (10 to 30), very strong (30 to 100), and extreme (>100) in favoring the alternative hypothesis; or anecdotal (BF10 = 1 to 0.33), moderate (0.33 to 0.1), strong (0.1 to 0.03), very strong (0.03 to 0.01) and extreme (<0.01) in favoring the null hypothesis (Lee and Wagenmakers’ classification) [36]. To calculate the probability of replicating the same results, we divided the actual BF10 value by BF10 + 1. The BF analysis was carried out through JAMOVI^®^.

Machine-learning random forest clustering analysis was applied aiming to stratify the sample according to the heart autonomic parameters. We included only variables that exhibited a moderate-to-extreme probability favoring the alternative hypothesis in Bayesian analysis. The lower BIC (Bayesian information criterion) was used to define the number of generated clusters and all machine-learning approaches were conducted in the software JASP (Version 0.17.2). After assigning each volunteer into the two generated clusters, we calculated the relative risk of subjects from each cluster to be an HS or HU patient.

## 3. Results

Comparison between the HS and HU groups showed a significant difference for the LFnu and HFnu spectral bands, as well as for the nonlinear parameter MSE values on scales from 3 to 20, and MSE_1–20_, MSE_1–5_, and MSE_6–20_ (*p* < 0.05). The HS group exhibited significantly higher entropy values in the cited scales when compared to the HU group (Table 2).

The Bayesian inference indicated a moderate probability (~66%) in favor of the alternative hypothesis for spectral parameters (LFnu and HFnu) in comparisons between HS vs. HU CKD patients. For MSE parameters, MSE1 exhibited an anecdotal probability in favor of the null hypothesis, while MSE2 presented a moderate probability (62.6%) in favor of the alternative hypothesis. MSE3 to MSE20 exhibited a moderate-to-very strong probability (79.4 to 96.3%) in favor of the alternative hypothesis. MSE_1–20_, MSE_1–5_, and MSE_6–20_ also exhibited a moderate-to-strong probability (88.0 to 92.8%) in favor of the alternative hypothesis.

All MSE parameters, except MSE1 and MSE2, achieved the criteria to be included in the machine-learning random forest clustering analysis, which generated two clusters. Twenty-eight volunteers were assigned to Cluster 1, which exhibited a greater heart-rate complexity, since all of the included MSE parameters exhibited greater values than observed in volunteers assigned to Cluster 2 (See Table 3).

Figure 1 exhibits the cluster density plots with the features of each MSE parameter from Clusters 1 and 2.

A 2 vs. 2 table analysis indicated that 81.0% of HS patients were also assigned to Cluster 1 (see Table 4), while subjects from Cluster 2 exhibited a relative risk = 2.94 [CI 95% = 1.14 to 7.58] (*p* = 0.025) of being in the HU group during a hemodialysis session.

## 4. Discussion

This study aimed to compare the variability of successive RR intervals between HS and HU CKD patients during a hemodialysis session, through linear and nonlinear methods. Our results showed that the HS group presents a greater contribution from the LFnu spectral band and a smaller contribution from the HFnu spectral band. Additionally, the entropy of the RR interval variability was significantly higher at Scales 3 to 20, as well as in MSE_1–20_, MSE_1–5_, and MSE_6–20_ during the hemodialysis session.

The contribution of the LFnu spectral band is expected to increase during hemodialysis sessions, since this is known to be associated with sympathetic activity [38]. Previous studies have already indicated that the inability to increase sympathetic activity during an acute blood volume decrease may predispose a patient to hypotension, which is the most frequent complication during hemodialysis sessions [8,9,20,39,40].

Our results confirmed the higher LFnu band in HS patients (mean difference [HS minus HU] and its 95% CI = 11.61 (0.90 to 22.33)); however, the Bayesian inference analysis indicated only a moderate posterior probability (BF = 1.93, probability = 65.9%) of replicating this result. This fact may be related to the limitation of linear methods in extracting information from datasets with high complexity, such as the RR interval variability obtained from CKD patients during the hemodialysis session, as we have already indicated previously [25].

Indeed, Silva et al. [25] demonstrated that nonlinear parameters obtained from the analysis of RR interval variability have a better reliability than linear parameters when the data are obtained from CKD patients during hemodialysis sessions. Data from other samples have also confirmed this statement [13]. In this context, the entropy analysis of successive RR intervals has been shown to be an excellent predictor of autonomic disorders [41], being more sensitive than the time and frequency domain parameters in extracting the level of system complexity/adaptation [42].

Nonlinear analysis methods are proposed as the better approach to describing the dynamics and complexity of biological phenomena [19,43,44]. Multiscale entropy (MSE) is a nonlinear method used to estimate the irregularity or unpredictability of a time series, with low entropy values indicating regular or predictable dynamics of an analyzed dataset. In contrast, high entropy values mean the opposite, and high entropy values are expected in complex systems with a high adaptive capacity [19,42]. Based on this concept, the high entropy values observed in HS patients from our study suggest better regulatory mechanism performance during hemodialysis sessions (i.e., better adaptability to volemic changes); in fact, complex systems are considered healthier [30].

It is important to emphasize that all frequentist inference was performed by adjusting the filtered volume during the hemodialysis session, since this variable is extremely important in defining the removed volume from the circulatory system; consequently, this is one of the main factors that predispose patients to IH.

Recent studies have shown the usefulness of applying MSE to assess the variability of RR intervals [30,45,46,47]. Specifically, Scales 1 to 5 seem to carry information about parasympathetic modulation, while Scales 6 to 20 seem to predominantly carry information about sympathetic modulation [30,45,46,47]. Thus, our results indicate that HS patients have a greater sympathetic modulation during hemodialysis sessions, as they have higher entropy values on Scales 6 to 20. It is important to highlight that sympathetic modulation is essential for maintaining blood pressure in conditions of withdrawal volume in short time intervals, such as in hemodialysis sessions. Additionally, the greater entropy on Scales 3 to 5, as well as MSE_1–5_, indicates a greater vagal modulation in HS patients, suggesting greater cardioprotection in conditions of cardiovascular stress.

It is important to emphasize that Bayesian inference indicated a higher posterior probability of difference between HS vs. HU patients for the nonlinear parameters (i.e., MSE) than linear (i.e., spectral) parameters. This fact indicates that despite being proposed as an indirect measure of sympathovagal modulation, the spectral parameters may not be sensitive enough to detect small differences between HS and HU CKD patients; the nonlinear parameters used in our study were shown to be more sensitive for this purpose, corroborating previous studies that demonstrated the better adequacy of nonlinear methods in obtaining information from the cardiovascular system in different conditions and populations [13,14,25,48]. Despite being a nonlinear approach, specifically an entropy-based metric, FuzzyEn did not demonstrate suitability for distinguishing hemodynamically stable from unstable patients during a hemodialysis session.

The machine-learning approach applied in our study allowed for stratifying our sample into two clusters according to the features of selected variables. Based on the established criteria, MSE parameters were included in the machine-learning analysis and the generated Cluster 1 was composed of subjects with a greater heart-rate complexity, since Cluster 1 exhibited greater values of all of the included MSE parameters. In fact, 81.0% of HS patients were also assigned to Cluster 1, confirming the association between greater heart-rate complexity and hemodynamic stability during hemodialysis sessions. Additionally, subjects with Cluster 2 features exhibited a 2.94-times higher (relative risk = 2.94 [CI 95% = 1.14 to 7.58]) probability of being in the HU group during a hemodialysis session.

Future studies should apply multivariate predictive models including the MSE parameters to predict the trend of hemodynamic instability during hemodialysis sessions, enabling the adoption of preventive strategies in relation to IH.

## 5. Conclusions

The results of the present study indicate that HS CKD patients exhibit higher values of the LFnu spectral band and a higher entropy (i.e., MSE) of RR intervals when compared to HI patients. Additionally, MSE exhibited a greater posterior probability of identifying differences in the pattern of variability of RR intervals between HS and HU patients during hemodialysis sessions.

## Figures and Tables

**Figure 1 entropy-25-00883-f001:**
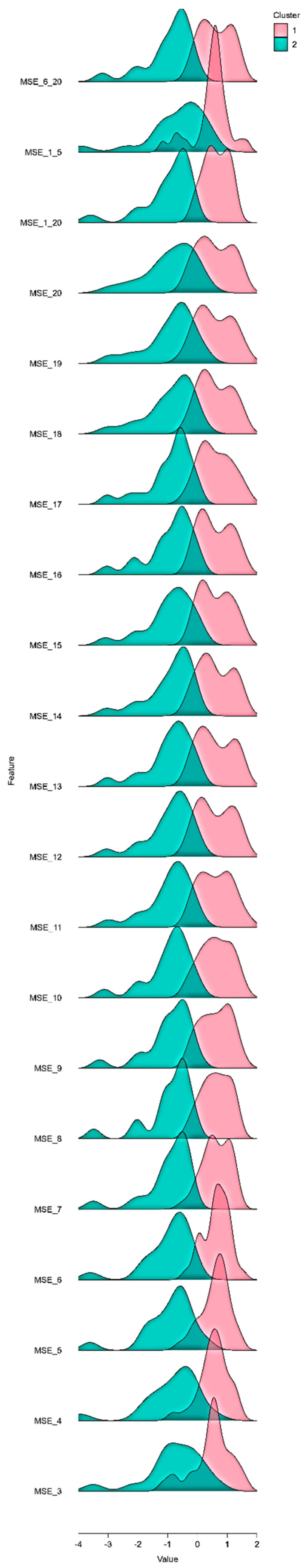
Cluster density plots presenting the features of each MSE parameter from Clusters 1 and 2.

**Table 1 entropy-25-00883-t001:** Sample characteristics and the filtered volume of the studied hemodialysis session.

Variable	All (n = 46)	HS (n = 21)	HU (n = 25)
Sex (M/F)	28/18	15/6	13/12
Age (years old)	49.0 ± 18.4	43.6 ± 19.3	53.6 ± 16.8
Dialysis time (months)	75.2 ± 71.6	63.9 ± 63.9	84.6 ± 77.4
SBP (mmHg)	149.0 ± 28.3	146.7 ± 23.8	151.0 ± 32.0
DBP (mmHg)	88.6 ± 17.5	87.7 ± 16.4	89.3 ± 18.7
Uremic state * (mg/dL)	133.2 ± 33.0	33.6 ± 7.3	31.9 ± 6.4
URR (%)	56.9 ± 14.7	15.5 ± 3.3	14.3 ± 2.9
Kt/V	2.3 ± 1.4	1.5 ± 0.3	1.4 ± 0.3
Filtered volume (mL)	2610.8 ± 955.0	1008.6 ± 220.1	919.6 ± 187.7

HS = hemodynamically stable; HU = hemodynamically unstable; SBP = systolic blood pressure; DBP = diastolic blood pressure; Kt/V = fractional urea clearance per dialysis; URR = urea reduction rate (*) mean from the last 12 months.

**Table 2 entropy-25-00883-t002:** Spectral (LFnu and HFnu) and nonlinear (MSE) parameters from successive RR intervals obtained during a hemodialysis session of hemodynamically stable vs. hemodynamically unstable CKD patients.

Variables	Groups		Mean Difference between Groups (CI 95%) [HS-HU]	*p* Value	BF_10,U_	Probability (%)
	HS (n = 21)	HU (n = 25)				
LFnu	72.6 ± 12.3	61.0 ± 21.6	11.61 (0.90 to 22.33) *	0.03	1.93	65.9
HFnu	27.3 ± 12.3	38.8 ± 21.5	−11.55 (−22.23 to −0.88) *	0.03	1.92	65.8
FuzzyEn	1.09 ± 0.31	1.12 ± 0.32	−0.035 (−0.22 to 0.15)	0.70	0.31	23.4
MSE 1	1.52 ± 0.21	1.46 ± 0.28	0.04 (−0.10 to 0.19)	0.55	0.37	27.2
MSE 2	1.64 ± 0.16	1.50 ± 0.26	0.13 (0.00 to 0.26)	0.05	1.67	62.6
MSE 3	1.58 ± 0.15	1.42 ± 0.24	0.17 (0.05 to 0.29) *	0.006	3.86	79.4
MSE 4	1.56 ± 0.16	1.34 ± 0.27	0.23 (0.10 to 0.36) *	0.001	15.8	94.0
MSE 5	1.55 ± 0.15	1.33 ± 0.26	0.24 (0.11 to 0.37) *	0.001	25.9	96.3
MSE 6	1.51 ± 0.17	1.30 ± 0.29	0.23 (0.09 to 0.37) *	0.002	8.13	89.0
MSE 7	1.45 ± 0.18	1.23 ± 0.30	0.25 (0.11 to 0.39) *	0.001	10.1	91.0
MSE 8	1.39 ± 0.19	1.16 ± 0.31	0.26 (0.11 to 0.41) *	0.001	9.79	90.7
MSE 9	1.31 ± 0.19	1.08 ± 0.31	0.25 (0.10 to 0.41) *	0.001	6.42	86.5
MSE 10	1.22 ± 0.21	1.01 ± 0.30	0.25 (0.10 to 0.40) *	0.002	5.39	84.4
MSE 11	1.16 ± 0.20	0.93 ± 0.31	0.26 (0.11 to 0.41) *	0.001	7.31	88.0
MSE 12	1.08 ± 0.20	0.87 ± 0.28	0.24 (0.10 to 0.38) *	0.002	6.27	86.2
MSE 13	1.00 ± 0.19	0.79 ± 0.25	0.23 (0.10 to 0.36) *	0.001	10.00	90.9
MSE 14	0.92 ± 0.17	0.75 ± 0.25	0.20 (0.07 to 0.33) *	0.003	4.6	82.1
MSE 15	0.88 ± 0.17	0.71 ± 0.24	0.20 (0.08 to 0.32) *	0.002	6.11	85.9
MSE 16	0.84 ± 0.16	0.67 ± 0.24	0.19 (0.07 to 0.31) *	0.002	5.2	83.9
MSE 17	0.80 ± 0.15	0.63 ± 0.22	0.19 (0.08 to 0.31) *	0.001	7.34	88.0
MSE 18	0.75 ± 0.14	0.59 ± 0.22	0.17 (0.06 to 0.28) *	0.003	5.00	83.3
MSE 19	0.71 ± 0.14	0.56 ± 0.22	0.17 (0.06 to 0.28) *	0.003	5.65	85.0
MSE 20	0.67 ± 0.13	0.53 ± 0.21	0.16 (0.05 to 0.26) *	0.004	3.62	78.4
MSE_1–20_	22.43 ± 2.70	18.88 ± 4.53	4.02 (1.82 to 6.21) *	0.001	12.8	92.8
MSE_1–5_	6.30 ± 0.51	5.66 ± 0.92	0.66 (0.21 to 1.10) *	0.005	6.63	86.9
MSE_6–20_	14.60 ± 2.38	11.91 ± 3.66	3.08 (1.27 to 4.89) *	0.001	7.34	88.0

Data are reported as the mean ± SD; (*) Significantly different (*p* < 0.05).

**Table 3 entropy-25-00883-t003:** Descriptive data of the included MSE parameters in the analysis to generate Clusters 1 (n = 28) and 2 (n = 18).

Variables	Clusters	
	Cluster 1 (n = 28)	Cluster 2 (n = 18)
MSE 3	1.60 ± 0.14	1.33 ± 0.21
MSE 4	1.57 ± 0.12	1.23 ± 0.24
MSE 5	1.58 ± 0.11	1.21 ± 0.21
MSE 6	1.56 ± 0.11	1.14 ± 0.22
MSE 7	1.50 ± 0.12	1.06 ± 0.22
MSE 8	1.44 ± 0.13	0.99 ± 0.24
MSE 9	1.36 ± 0.15	0.91 ± 0.23
MSE 10	1.28 ± 0.15	0.84 ± 0.22
MSE 11	1.21 ± 0.16	0.77 ± 0.22
MSE 12	1.12 ± 0.15	0.72 ± 0.21
MSE 13	1.04 ± 0.14	0.66 ± 0.19
MSE 14	0.97 ± 0.13	0.61 ± 0.19
MSE 15	0.92 ± 0.12	0.57 ± 0.19
MSE 16	0.88 ± 0.12	0.54 ± 0.18
MSE 17	0.83 ± 0.12	0.51 ± 0.17
MSE 18	0.78 ± 0.11	0.48 ± 0.17
MSE 19	0.74 ± 0.12	0.45 ± 0.16
MSE 20	0.70 ± 0.11	0.42 ± 0.16
MSE_1–20_	23.12 ± 1.73	16.42 ± 3.51
MSE_1–5_	6.34 ± 0.50	5.35 ± 0.87
MSE_6–20_	15.21 ± 1.76	9.90 ± 2.70

Data are reported as the mean ± SD.

**Table 4 entropy-25-00883-t004:** Absolute and relative frequency of CKD patients assigned to Clusters 1 (n = 28) and 2 (n = 18) according to the hemodynamic stability during a hemodialysis session.

		Groups	
		HS (n = 21)	HU (n = 25)
Clusters	Cluster 1 (n = 28)	17 (81.0%)	11 (44.0%)
	Cluster 2 (n = 18)	4 (19.0%)	14 (56.0%)

## Data Availability

The data presented in this study are available on request from the corresponding author.

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
