# Peer review of "Comparison of Heart Autonomic Control between Hemodynamically Stable and Unstable Patients during Hemodialysis Sessions: A Bayesian Approach"

_entropy, 2023, doi:10.3390/e25060883_

Round 1
Reviewer 1 Report
The subject of the paper is interesting and so are the results. The authors show the difference in heart autonomic control between stable and unstable patients during hemodialysis session.
For this purpose they employed linear and non-linear HRV metrics. From the linear metrics they chose LFnu and HFnu. From the non-linear ones, multi-scale entropy. Even though multi-scale entropy produces estimators in many scales, the number of the metrics used is very small (2+1). I would expect, at least the popular metrics from the time domain and the popular definitions of entropy (Shannon, approximate, sample) to be included. Since the subject of the journal is entropy, I would also expect the main analysis to be performed with entropy measures, so I would also include less popular indices like Renyi entropy, permutation entropy, fuzzy entropy, bubble entropy, etc. I do not expect the authors to include all those these metrics in their analysis, but I would expect to increase the number of entropy based metrics.
In addition, each metric is evaluated independently. There is an exception with some scale regions of multi-scale entropy, where regions and not single scales are considered. A more complex analysis combining more than one metric would be interesting.
One way to do it and very trending is machine learning classification. Authors can consider employing at least one simple classifier to test how well the two groups of subjects are discriminated. Machine learning classifiers combines all features fed as input and detect deeper and non-linear relations between subjects, leading to more interesting results.
Author Response
Dear reviewer,
Thank you for the time and effort you put forth in reviewing our submission. We are submitting a revised version of the manuscript “COMPARISON OF HEART AUTONOMIC CONTROL BETWEEN HEMODYNAMICALLY STABLE AND UNSTABLE PATIENTS DURING HEMODIALYSIS SESSION: AN BAYESIAN APPROACH” that was amended in response to the criticism of the reviewers. We accepted all your criticism and revised the manuscript accordingly. Below, we answer each of the comments.
The subject of the paper is interesting and so are the results. The authors show the difference in heart autonomic control between stable and unstable patients during hemodialysis session.
For this purpose they employed linear and non-linear HRV metrics. From the linear metrics they chose LFnu and HFnu. From the non-linear ones, multi-scale entropy. Even though multi-scale entropy produces estimators in many scales, the number of the metrics used is very small (2+1). I would expect, at least the popular metrics from the time domain and the popular definitions of entropy (Shannon, approximate, sample) to be included. Since the subject of the journal is entropy, I would also expect the main analysis to be performed with entropy measures, so I would also include less popular indices like Renyi entropy, permutation entropy, fuzzy entropy, bubble entropy, etc. I do not expect the authors to include all those these metrics in their analysis, but I would expect to increase the number of entropy based metrics.
Answer:
Dear reviewer, thank you for your kind comments. We added the analysis with Fuzzy entropy as requested.
In addition, each metric is evaluated independently. There is an exception with some scale regions of multi-scale entropy, where regions and not single scales are considered. A more complex analysis combining more than one metric would be interesting.
One way to do it and very trending is machine learning classification. Authors can consider employing at least one simple classifier to test how well the two groups of subjects are discriminated. Machine learning classifiers combines all features fed as input and detect deeper and non-linear relations between subjects, leading to more interesting results.
Answer:
Dear reviewer, thank you for the suggestion. We added a machine-learning method as requested.
Reviewer 2 Report
In general, this is an interesting manuscript depicting the power of MSE in the statistical investigation of heart autonomic control during hemodialysis session.
Few issues should be sorted out before this manuscript could be recommended for publication in the Journal.
The authors choose MSE as the statistical indicator. It would be truly beneficial if the authors could present (at least discuss) the alternative approaches. Otherwise, the selection of MSE may look like a matter of luck. In other words, a comparison with other algorithms would enrich the presentation.
The computation of MSE is performed using the time delay for different scales. The authors should at least comment the effects induced by different scaling factors, and not only:
The discriminant statistic based on MPE-MWPE relationship and non-uniform embedding, Journal of Measurements in Engineering, Vol. 10, No. 3, pp. 150–163, Sep. 2022, https://doi.org/10.21595/jme.2022.22897
All measurements and computations are performed on human ECGs (RR intervals). What about the ethics permit?
A minor revision is recommended.
Minor improvements in the presentation style are recommended.
Author Response
Dear reviewer,
Thank you for the time and effort you put forth in reviewing our submission. We are submitting a revised version of the manuscript “COMPARISON OF HEART AUTONOMIC CONTROL BETWEEN HEMODYNAMICALLY STABLE AND UNSTABLE PATIENTS DURING HEMODIALYSIS SESSION: AN BAYESIAN APPROACH” that was amended in response to the criticism of the reviewers. We accepted all your criticism and revised the manuscript accordingly. Below, we answer each of the comments.
In general, this is an interesting manuscript depicting the power of MSE in the statistical investigation of heart autonomic control during hemodialysis session.
Few issues should be sorted out before this manuscript could be recommended for publication in the Journal.
The authors choose MSE as the statistical indicator. It would be truly beneficial if the authors could present (at least discuss) the alternative approaches. Otherwise, the selection of MSE may look like a matter of luck. In other words, a comparison with other algorithms would enrich the presentation.
The computation of MSE is performed using the time delay for different scales. The authors should at least comment the effects induced by different scaling factors, and not only:
The discriminant statistic based on MPE-MWPE relationship and non-uniform embedding, Journal of Measurements in Engineering, Vol. 10, No. 3, pp. 150–163, Sep. 2022, https://doi.org/10.21595/jme.2022.22897
Answer:
Dear reviewer, thank you for the suggestion. We added the analysis with Fuzzy entropy and a machine-learning analysis method as requested by another reviewer. We think this approach reinforced the relevance of entropy measures, especially from the MSE.
All measurements and computations are performed on human ECGs (RR intervals). What about the ethics permit?
Answer:
Dear reviewer, all procedures used in our study were evaluated and approved by a Research Ethics Committee. Additionally, all invited volunteers to take part in our study were informed about the study procedures. We presented the ethical aspects in our manuscript:
“Volunteers were invited to take part in the study and were informed about the study procedures; those who agreed signed informed consent. The study was approved by the Research Ethics Committee of the Universidade Estadual do Sudoeste da Bahia (protocol#: 09635912.3.0000.0055).”
Round 2
Reviewer 1 Report
I would like to thank the authors for their effort. I believe the manuscript is at a good scientific level